# Evaluation of participant reluctance, confidence, and self-reported behaviors since being trained in a pharmacy Mental Health First Aid initiative

Matthew Witry[ID][1]*, Hacer Karamese[2], Anthony Pudlo[3]

**1** Division of Health Services Research, Department of Pharmacy Practice and Science, University of Iowa College of Pharmacy, Iowa City, Iowa, United States of America, **2** Center for Evaluation and Assessment, University of Iowa College of Education, Iowa City, Iowa, United States of America, **3** Iowa Pharmacy Association, Des Moines, Iowa, United States of America

* matthew-witry@uiowa.edu

**Data Availability Statement:** Data are available via figshare 10.6084/m9.figshare.11914872.

**Funding:** MW: Community pharmacy foundation grant #213. www.communitypharmacyfoundation.

## Abstract

In the U.S., an estimated one in five individuals experience a mental illness annually which contribute to significant human and economic cost. Pharmacists serving in a public health capacity are positioned to provide first aid level intervention to people experiencing a mental health crisis. Research on pharmacy professionals (pharmacists, technicians, students) undergoing training in Mental Health First Aid (MHFA) can provide evidence of the potential benefits of such training. The objectives of this study were to 1) describe the reluctance and confidence to intervene in mental health crises of pharmacy professionals previously trained in MHFA, 2) describe their self-reported use of MHFA behaviors since becoming trained, and 3) describe participant open-ended feedback on their MHFA training. Materials and methods: An electronic survey was disseminated in May and June, 2019 using a four-email sequence to pharmacy professionals who had completed MHFA training from one of five pharmacist MHFA trainers throughout 2018. Domains included demographics, six Likert-type reluctance items, seven Likert-type confidence items for performing MHFA skills, and frequency of using a set of nine MHFA skills since being trained. Prompts collected open-ended feedback related to MHFA experiences and training. Descriptive statistics were used for scaled and multiple-choice items and a basic content analysis was performed on the open-ended items to group them into similar topics. Results: Ninety-eight out of 227 participants responded to the survey yielding a response rate of 44%. Participants reported high levels of disagreement to a set of reluctance items for intervening and overall high levels of confidence in performing a range of MHFA skills. Participant self-reported use of a set of MHFA skills ranged from 19% to 82% since being trained in MHFA. Almost half (44%) of participants had asked someone if they were considering suicide. A majority (61%) also had referred someone to resources because of a mental health crisis. Open-ended responses included positive experiences alongside important challenges to using MHFA in practice and recommendations including additional training focused on the pharmacy setting.

org/ The funders had no role in study design, data
collection and analysis, decision to publish, or
preparation of the manuscript.

**Competing interests:** AP was one of the
pharmacist MHFA facilitators. This does not alter
our adherence to PLOS ONE policies on sharing
data and materials.

Conclusions: Pharmacy professionals in this evaluation reported little reluctance and high
confidence related to using MHFA training and reported use of MHFA skills since being
trained.

## Introduction

It is estimated that one in five U.S. adults experience a mental illness annually but fewer than
half receive treatment in a given year. [1] Untreated and undertreated mental illness also has a
significant economic cost with an estimated 193.2 billion in lost productivity every year. [1] A
large portion of the U.S. has a deficiency of mental health service providers and significant
gaps in access to mental health services. [2]

Because of the high prevalence of under treated mental illness, there has been a public
health effort to train non-mental health professionals to provide first aid level care to people
experiencing mental illnesses and mental health crises. [3–5] Mental Health First Aid is one
such training and is supported by the National Council on Behavioral health. The goal of
MHFA is to train community members to decrease stigma toward mental illness, serve as a
"first line of support" or gatekeepers for distressed people, and help people in crisis seek fur-
ther assistance. [6] Mental Health First Aid U.S.A.is based on a five-step action plan that uses
the acronym ALGEE$^{®}$: assess for risk of suicide or self-harm, listen non-judgmentally, give
reassurance and information, encourage appropriate professional help, and encourage self-
help and other support strategies. [6]

Pharmacists are accessible and trusted health care professionals [7] that potentially can
intervene and refer these individuals to help. [4, 8–11] Persons experiencing mental health cri-
ses, including those with warning signs of suicide, can present to pharmacies, clinic appoint-
ments, or other pharmacy practice settings during routine interactions such as when people
procure medications, seek advice over the telephone, or engage with other pharmacy services.
[8,10–14] Studies in pharmacy, however, cite a variety of barriers to intervening [8, 10, 15, 16],
including a lack of confidence by the pharmacist [14, 17] and a need for additional training.
[13,17] Pharmacists, especially those in the community-based setting, could play a key role in
identifying people in mental health crises and refer them to appropriate professional help. [10,
11, 14, 18, 19]

Research suggests Mental Health First Aid (MHFA) can increase trainee knowledge,
decrease reluctance, decrease stigma, and improve confidence related to intervening with per-
sons experiencing mental health crises, including suicidal ideation. [20,21] Studies also suggest
that MHFA training increases behaviors like identifying people in mental health crises and
making referrals in the months and years following the initial MHFA training. [21–23]

Formal evaluations of MHFA and other gatekeeper training programs for pharmacy train-
ees beyond the initial training are lacking. [24, 25] Educational interventions related to mental
health crisis intervention during pharmacy school and offered as continuing education for
practicing pharmacists and pharmacy technicians may have an impact on improving confi-
dence and knowledge and decreasing stigma, [3, 12, 24, 26–31] but more research is needed
on how often trainees use gatekeeper skills in practice and their feelings of competency months
and years after their initial training. Research on the longitudinal experience of pharmacy pro-
fessionals that have been trained as mental health gatekeepers through programs like MHFA
may provide evidence of program effectiveness and impact. The objectives of this research are
to describe: 1) the reluctance and confidence of pharmacy professionals previously trained in

MHFA, 2) self-reported use of MHFA behaviors since becoming trained, and 3) participant open-ended feedback on their MHFA training.

## Materials and methods

This evaluation was based on an initiative by the National Community Pharmacy Association, Iowa Pharmacy Association, and the Community Pharmacy Foundation where funding was provided for 5 pharmacists to become accredited by the National Council on Behavioral Health to conduct MHFA U.S.A. training sessions. The initial week-long training of these 5 pharmacists occurred in 2017 or 2018. These trainers then facilitated MHFA training events for over 200 pharmacists, pharmacy faculty, pharmacy students, and pharmacy technicians in 2018. The MHFA training sessions were offered through a state pharmacy association to local individuals via an online signup, to student pharmacist groups, and to workshop attendees. Pharmacists attending any of the MHFA training sessions could receive eight hours of continuing education credit for their participation. This study was approved by the University of Iowa Institutional Review Board on May 8, 2019 IRB #201905722.

MHFA is an eight-hour training that introduces non-mental health professionals and the lay public to a five-step action plan for assessing individuals and making referrals for people experiencing mental health crises or illnesses. [6] The MHFA training sessions were delivered in-person either over a single 8-hour session or two 4-hour sessions to groups in Iowa, California, Florida, Indiana, and Oklahoma.

This study was guided by evaluation standards and used a single group post-only design. [27] Data were collected using an anonymous electronic survey administered through Qualtrics (Provo, UT). A sequence of four email contacts was used which include a pre-notification email, initial email, and two reminder emails. People could email the research team if they wanted to opt out of future emails. Each of the four emails was sent approximately six to eight days apart in order to stagger the day of the week that people were receiving emails. Four of the five pharmacist trainers provided email lists for the participants they had trained using the email addresses participants submit as part of their certification. One pharmacist was not used to supply participant lists as they provided MHFA in conjunction with one of the other trainers. Emails were sent between May and June, 2019 which was approximately 6–18 months following the trainings.

The survey included domains informed by a gatekeeper evaluation model by Burnette et al. [3] The domains included demographics, reluctance, confidence, and use of MHFA behaviors. Demographics included age, self-identified gender, and professional role. Reluctance was assessed using six items associated with reluctance to engage with mental health crises and were based on previous gatekeeper research. [26,32] Reluctance items included suicide and mental health misconceptions "If someone contemplating suicide does not seek assistance, there is nothing I can do to help" and environmental factors like "I am too busy to provide mental health first aid at work." Confidence in performing MHFA skills was assessed using seven items and based on established MHFA self-evaluation questions. [6, 33] Confidence items focused on the ALGEE behaviors and included "Ask someone if they are thinking about suicide" and "Encourage someone experiencing a mental health crisis to seek professional help." Use of MHFA behaviors was assessed using 9-items based on the MHFA training and other studies. [6, 22, 33] For these self-reported behaviors, participants selected if they had done them 0,1,2,3, 4 or more times, or not applicable. The self-reported behaviors included both thoughts like "Thought someone's behavior might indicate they are having suicidal thoughts" and behaviors like "Asked someone if they are considering suicide" and "Referred someone to appropriate resources because you were concerned they were considering suicide."

Both thoughts and behaviors were assessed in the event that behaviors were rare. A median number of times was calculated by treating 4 or more as 4. Median and interquartile ranges were calculated for scaled items that demonstrated a non-normal distribution

Six open-ended text boxes were available for participants to provide free text responses. These included: Without using any identifying information, do you have any experience with MHFA you wish to share?; What major challenges or problems have you faced while using MHFA in your role as a pharmacist? What improvements would you recommend for future MHFA training recruitment? What suggestions would you propose related to how MHFA training is delivered? What suggestions would you propose for further supporting pharmacists in the area of mental healthcare? What do you still need help with? and Is there anything else on this topic not included in the previous questions that you feel important to tell us?

The survey was not piloted on the participants because we did not want to decrease the people available to take the survey. Multiple rounds of feedback were obtained from outside the research team using experts in evaluation and suicidology. Survey questions were phrased generally as opposed to referring specifically to the respondent's professional role because as a gateleeper training, the goal is to give people skills to use regardless of the setting.

Descriptive statistics including frequencies, medians, and percents were calculated for all scaled items and frequencies and percents were calculated for multiple choice multiple-choice responses. Reliability coefficients were calculated for the confidence and reluctance scales. Open-ended responses were analyzed using a basic descriptive content analysis. [34] This included applying codes to describe text segments from the open-ended responses in an interactive process to group similar ideas from the participants. One author conducted an initial sorting and another author provided feedback. Discrepancies were resolved through discussion.

## Results

Ninety-eight out of 227 invited participants responded to the survey which participants completed between 6 and 18 months since their training. Four emails were undeliverable, yielding a usable response rate of 44%. The median survey-completion time was 7 minutes. Sixty-two percent of respondents were female. Almost half of respondents were practicing pharmacists (46%) and 23% were student pharmacists (Table 1). Half of participants were trained between January and June, 2018.

Responses to the reluctance items favored disagreement suggesting respondents were willing to intervene with people experiencing a mental health crisis (Table 2). Participants most frequently reported strong disagreement to the statement that "I am too busy to provide mental health first aid at work (N = 57)."

Responses to positively worded confidence items favored positive answers with 5 of 7 item frequencies demonstrating strong agreement with the specific MHFA skill (Table 3). The greatest number of respondents strongly agreed they could listen non-judgmentally (n = 70) and encourage someone experiencing a mental health crisis to seek help (n = 64).

Participant self-reported use of MHFA skills ranged from 19% to 82% using the skill since being trained in MHFA (Table 4). Most had asked someone about their distressed mood (82%), with 28% doing so 4 or more times. Almost half (44%) of participants had asked someone if they were considering suicide. A majority (61%) also had referred someone to resources because they were concerned they might be experiencing a mental health crisis.

The responses to the open-ended questions indicated positive MHFA experiences, challenges to using MHFA in practice, and insights for training improvements. For MHFA

Table 1. Demographics of survey respondents who participated in MHFA training (n = 98).

| Variable | Frequency | Percentage |
|---|---|---|
| Gender | | |
| Female | 61 | 62% |
| Male | 22 | 22% |
| Non-binary gender | 1 | 1% |
| Age category | | |
| Age below 25 | 9 | 9% |
| Age 25–34 | 32 | 33% |
| Age 35–44 | 20 | 20% |
| Age 45–54 | 8 | 8% |
| Age 55–64 | 13 | 13% |
| Age above 65 | 2 | 2% |
| Pharmacy role | | |
| Practicing pharmacist | 45 | 46% |
| Student pharmacist | 23 | 23% |
| Pharmacy school faculty | 6 | 6% |
| Pharmacy technician | 2 | 2% |
| Training period | | |
| Trained January-March, 2018 | 16 | 16% |
| Trained April-June, 2018 | 34 | 35% |
| Trained July-September, 2018 | 27 | 28% |
| Trained October-December, 2018 | 20 | 20% |

Demographics not completed for n = 14.

experiences, 10 shared their success stories in using MHFA skills in practice and 3 shared positive thoughts on the training.

> "*I talked to a patient on the phone who, I believe, was having a panic attack. She had questions about her new medication and asked several times if she was going to die. I recognized the feeling of dying as one of the signs of a panic attack and reassured her she was ok and talked her through it. She seemed to get better as I talked to her and seemed perfectly fine the next day.*"

> "*A very positive experience. Some of the dialog examples have come in handy in a multitude of situations, including with both employees and patients.*"

Twenty-two participants shared challenges or problems they have faced while using MHFA. Eleven participants described time constraints as a major challenge for using MHFA in practice. Participants stated that they do not have enough time given their workload to interact with people to determine the nature of their crisis, provide listening and support, and create and execute an intervention plan.

> "*[A busy community pharmacy setting] is not at all an ideal environment to even quickly assess if patients may need further counseling or reassurance of their symptoms. If the line is long, you will have the manager and even other customers in line irate and pushy to get that patient in front of you out of the door as soon as possible.*"

**Table 2. Responses to reluctance items for engaging in a mental health crisis (N = 98).**

| Item | Percent response | | | | | | | Median (IQR) |
|---|---|---|---|---|---|---|---|---|
| | 1 = SD | 2 = D | 3 = N | 4 = A | 5 = SA | NA | NR | |
| There is very little that I can do to help if someone thinking about suicide doesn't acknowledge the situation | 34 | 40 | 8 | 7 | 3 | 0 | 8 | 2 (1) |
| If someone contemplating suicide does not seek assistance, there is nothing I can do to help | 48 | 34 | 5 | 3 | 2 | 0 | 8 | 1 (1) |
| If someone in a mental health crisis refuses to seek help, it should not be forced upon then | 18 | 40 | 15 | 11 | 6 | 0 | 9 | 2 (1) |
| I cannot understand why anyone would contemplate suicide | 56 | 18 | 12 | 1 | 3 | 1 | 8 | 1 (1) |
| I am too busy to provide mental health first aid at work | 58 | 19 | 7 | 4 | 3 | 0 | 8 | 1 (1) |
| I do not know most patients well enough to know when they are in a mental health crisis | 27 | 29 | 15 | 9 | 6 | 6 | 8 | 2 (2) |

SD = Strongly disagree, D = Somewhat disagree, N = Neither agree or disagree, A = Somewhat agree, SA = Strongly agree, NA = Not Applicable, NR = No Response, IQR = Interquartile range

Cronbach's Alpha = 0.79

While time constraints were reported as a challenge, some participants were still motivated to help people. For example, one participant stated that time "is a factor" but "not an excuse for not stepping up when needed." Similarly, another participant indicated that helping people in the pharmacy requires "a lot of time" but it is "worth it." Four participants indicated not having enough skills and resources as a challenge. One participant stated, "Realizing I still have a lot to learn, that people react to crisis situations in so many different ways." Another participant said it is "a challenge to remember what was taught" and another found it challenging to differentiate a mental health crisis from "a bad day."

Other challenges related to better understanding local resources and "understanding the system" and a lack of high-quality mental health providers in the area, including challenges with insurance coverage. Respondents also voiced concerns about privacy for these discussions and not having enough background information about their patients hampering their abilities to effectively intervene. Three respondents also saw stigma as a barrier, including other staff which may have stigmatizing beliefs about people with mental illness or self-stigmatizing beliefs held by people that may keep them from approaching the pharmacist.

The participants also shared insights on ways to improve the training and recruitment efforts. Ten participants recommended training more people, across settings and positions.

**Table 3. Responses to confidence items for performing Mental Health First Aid skills (N = 98).**

| Item: I am confident I can. . . . | Percent response | | | | | | Median (IQR) |
|---|---|---|---|---|---|---|---|
| | 1 = SD | 2 = D | 3 = N | 4 = A | 5 = SA | NR | |
| Recognize the signs that someone may need MHFA | 1 | 1 | 0 | 46 | 39 | 11 | 4 (1) |
| Ask someone if they are thinking about suicide | 0 | 0 | 2 | 46 | 39 | 11 | 4 (1) |
| Listen non-judgmentally to someone experiencing a mental health crisis | 0 | 0 | 2 | 15 | 70 | 11 | 5 (0) |
| Offer basic "first aid" level information to someone experiencing a mental health crisis | 0 | 0 | 1 | 32 | 54 | 11 | 5 (1) |
| Offer reassurance to someone experiencing a mental health crisis | 0 | 0 | 0 | 35 | 52 | 11 | 5 (1) |
| Encourage someone experiencing a mental health crisis to seek professional help | 0 | 0 | 0 | 23 | 64 | 11 | 5 (1) |
| Encourage self-help strategies for someone experiencing a mental health crisis | 0 | 2 | 3 | 33 | 49 | 11 | 5 (1) |

SD = Strongly disagree, D = Somewhat disagree, N = Neither agree or disagree, A = Somewhat agree, SA = Strongly agree, NA = Not Applicable, NR = No Response, IQR = Interquartile range

Cronbach's Alpha = 0.82

**Table 4. Respondent self-reported use of Mental Health First Aid skills since training (N = 98).**

| | Percent response | | | | |
|---|---|---|---|---|---|
| | No | Yes | NA | NR | Median times if Yes (IQR) |
| Thought someone's behavior might indicate they are having a mental health crisis | 13 | 72 | 1 | 13 | 2 (1) |
| Thought someone's behavior might indicate they are having suicidal thoughts | 28 | 57 | 2 | 13 | 2 (1) |
| Asked someone about their distressed mood | 4 | 82 | 0 | 14 | 3 (2) |
| Asked someone if they are considering suicide | 40 | 44 | 2 | 14 | 2 (1) |
| Listened non-judgmentally to someone experiencing a mental health crisis | 4 | 80 | 2 | 14 | 3 (2) |
| Referred someone to appropriate resources because you were concerned they might be experiencing a mental health crisis | 22 | 61 | 2 | 14 | 2 (2) |
| Referred someone to appropriate resources because you were concerned they were considering suicide | 37 | 45 | 3 | 15 | 1 (1) |
| Engaged with a mental health crisis resource on behalf of someone | 51 | 32 | 3 | 14 | 2 (2) |
| Engaged with emergency medical or police services because of someone experiencing a mental health crisis | 63 | 19 | 4 | 13 ( | 2 (2) |

NA = Not Applicable, NR = No Response median = if yes, how often? 1,2,3,4 or more times

"*all healthcare providers should be required to learn*"

There also was interest in having the MHFA training occur at work and while being paid. For the content of training, there was interest in making it more specific to the pharmacy setting, including how MHFA could be used in brief pharmacy encounters. Two participants indicated that the content of the training was "repetitive" and suggested there could be higher level courses. Two participants suggested providing additional resources, such as a "brief refresher", a phone app, and/or a card for "basic steps" that could be used as a reminder. Lastly, there was an interest in quality continuing education to maintain learning.

## Discussion

Overall, people completing the survey reported disagreement to items assessing their reluctance to engage in MHFA and high levels of confidence in performing a range of MHFA behaviors. These pharmacy participants also reported using a range of MHFA skills since their training, often on more than one occasion. This is among the first longitudinal evaluation of a MHFA or other gatekeeper training in pharmacy to include self-reported behaviors since being trained.

The largest group of respondents was practicing pharmacists followed by student pharmacists. The students that responded were nearing the end of their schooling and may be speaking from practice experience on clinical rotations or during part-time work. The sample had a large participation of younger respondents, but this can be attributed to some of the MHFA sessions being targeted to students through a pharmacy school. Exposing student pharmacists to mental health crisis intervention training may have benefits both in and out of their work settings. [11, 24, 28, 30]

Regarding participant reluctance, most provided responses that are considered conducive to intervening with people experiencing mental health crises. One of the strongest held beliefs was disagreement that "I am too busy to provide mental health first aid at work." While encouraging, this contradicts some of the open-ended responses and findings from other

research suggesting it is hard to engage in mental health crisis intervention in the community pharmacy setting. [14–16] These answers may be subject to social desirability bias and suggest an aspirational state. Alternatively, consistent with some of the open-ended responses, participants may be committed to make time should the situation warrant. Although, people may be less likely to engage if a pharmacist seems too busy, lacks an established relationship, or feels stigmatized at the pharmacy. [16, 35]

The reluctance item with the lowest score was "I do not know most patients well enough to know when they are in a mental health crisis." While responses, on average, were more favorable than neutral, this appears to be a potential barrier to consider. Pharmacies should continue to work to better understand the people that use their pharmacy and establish relationships as doing so can decrease stigma and increase help-seeking. [10, 15, 35]. While having a prior relationship has been reported by pharmacists as a facilitator for mental health crisis intervention [15] it should not be a requirement as MHFA is intended for any person, regardless of background, to help someone without needing a prior relationship. [6]

Respondents were highly confident in their abilities to use MHFA skills. They were most confident in their ability to listen non-judgmentally to someone experiencing a mental health crisis. Non-judgment is associated with decreased stigma, which MHFA evaluation studies have shown [20, 21], although stigma was not measured in the present study. This very high perception of listening non-judgmentally may be part an aspirational belief as remaining out of judgment can be difficult [36] and respondents could be over-estimating their abilities as this was one of the more subjective skills. There is evidence pharmacists tend toward medication-centric communication rather than patient-centric communication, including with mental illness. [37, 38]

Confidence also was high for encouraging someone experiencing a mental health crisis to seek professional help. We cannot be certain if participants interpreted this item as giving out the national suicide hotline number, encouraging someone to see a therapist, or something else. These overall high levels of confidence warrant testing as some research shows inconsistent performance of crisis intervention skills by pharmacists and students such as directly asking about suicide. [33, 39]

Encouragingly, respondents used a range of MHFA skills in the months following their initial training. Most participants had asked someone about their distressed mood and almost half reported asking someone if they were having suicidal thoughts and a similar percent referred someone to appropriate resources because of suicidal ideation. A caveat to these self-reported behaviors is that we do not know if these happened at work with a pharmacy patron, at work with a co-worker, or in someone's personal life outside of work. Either way, these interventions represent potentially life-saving interventions. These MHFA trained pharmacy professionals appeared to ask about suicide in greater numbers than a general pharmacist population where only 14% had asked about suicide and more commonly the information was volunteered. [14]

The open-ended responses offer several areas for improvement and future work related to training. First, there was interest in continuing education after initial training. This could take the form of a refresher course, or intervening in mental health crises in pharmacy-specific environments such as the community or retail setting as the standard MFHA training does not address the environmental constraints of a community pharmacy. Such trainings are needed as the initial MHFA training certification is only for 3 years and the recertification course focuses on reviewing basic concepts rather than adding new concepts.

There also was interest in additional resources like a pocket guide or other brief ways to stay engaged with the skills presented in the training. For example, Washington State has mandated suicide prevention training for pharmacists [40] and has developed materials. There are

other suicide focused trainings that pharmacists can take, such as question persuade refer (QPR) [41]. Some open-ended comments pointed out a perceived lack of local mental health professionals. Shortages of mental health professionals have been reported, especially in rural areas [2]. Pharmacy professionals also may need more training in navigating local mental health resources.

These data also raise several organizational concerns. Pharmacies should consider how to accommodate a pharmacist taking time out to interact with someone experiencing a mental health crisis. This could require considerations of staffing levels and private consultation areas [15, 16]. There also was a recommendation to have the MHFA training during work hours or on the clock. Employers should consider this as a benefit to employment and strengthening their workforce, including technicians and management.

This evaluation study has several limitations. With a single group design, it cannot be determined what proportion of participant beliefs and self-reported use of MHFA behaviors are attributable to their participation in one of the pharmacist-led MHFA training sessions. Experimental designs are needed. Also, it was not possible to evaluate differences among trainers. All surveys were disseminated at one time point and no adjustments were made based on when in 2018 the trainee completed their MHFA training. The survey data are based on self-report and the MHFA behaviors described in this report were not based on observation and may have variability in interpretation and may be subject to social desirability bias. There also could be response bias as most participants voluntarily pursued MHFA training and volunteered to take the evaluation survey. While the response rate was high for an electronic survey, non-responders may have had different experiences since being certified such as not having used MHFA in practice. While there were participants from multiple U.S. states, larger samples would be needed for greater confidence in generalizability.

This study also suggests several areas for future research. First, there is a need for larger randomized studies with pre and post evaluations, and longer term follow up. There also is a need to collect observational data to supplement self-report to determine if pharmacy professionals are effective in their mental health crisis interventions. Training topics that could supplement MHFA training could relate to restricting access to medications as a lethal means [9, 34] and counseling on the suicide risk associated with antidepressants. [42, 43] Calling for, and developing trainings and resources for making modifications to work environments are additional opportunities to improve how pharmacy professionals engage in mental health crisis intervention which would require further development and evaluation. Lastly, future work should consider measuring stigma and environmental factors as these have been identified as theoretically important to modeling mental health crisis interventions. [3]

## Conclusions

This study found pharmacy professionals trained by pharmacist MHFA trainers had low reluctance and high levels of confidence in using the range of skills taught in the 8-hour MHFA program. Respondents also reported engaging in a range of MHFA behaviors since being trained such as asking about suicide, referring someone to resources, and engaging with a mental health crisis resource on someone's behalf. While pharmacy professionals were positive about the training, there was interest in pharmacy-specific continuing education such as providing MHFA within realistic pharmacy environments.

## Supporting information

**S1 File.**
(DOCX)

## Author Contributions

**Conceptualization:** Matthew Witry, Anthony Pudlo.

**Data curation:** Matthew Witry, Anthony Pudlo.

**Formal analysis:** Matthew Witry, Hacer Karamese.

**Funding acquisition:** Matthew Witry, Anthony Pudlo.

**Investigation:** Matthew Witry, Anthony Pudlo.

**Methodology:** Matthew Witry, Hacer Karamese.

**Project administration:** Matthew Witry.

**Resources:** Matthew Witry.

**Software:** Hacer Karamese.

**Supervision:** Matthew Witry.

**Validation:** Matthew Witry.

**Writing – original draft:** Matthew Witry.

**Writing – review & editing:** Matthew Witry, Hacer Karamese, Anthony Pudlo.

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
