## [Decision Letter · Decision Letter 0]

13 Jan 2020

PONE-D-19-33742

Evaluation of Mental Health First Aid training initiative for pharmacy

PLOS ONE

Dear Dr Witry,

Thank you for submitting your manuscript to PLOS ONE. After careful consideration, we feel that it has merit but does not fully meet PLOS ONE’s publication criteria as it currently stands. Therefore, we invite you to submit a revised version of the manuscript that addresses the points raised during the review process.

Please provide a response to the included reviews. 

We would appreciate receiving your revised manuscript by Feb 27 2020 11:59PM. To enhance the reproducibility of your results, we recommend that if applicable you deposit your laboratory protocols in protocols.io, where a protocol can be assigned its own identifier (DOI) such that it can be cited independently in the future. For instructions see: http://journals.plos.org/plosone/s/submission-guidelines#loc-laboratory-protocols

We look forward to receiving your revised manuscript.

Kind regards,

Andrew Soundy

Academic Editor

PLOS ONE

Journal Requirements:

2. Please include additional information regarding the survey or questionnaire used in the study and ensure that you have provided sufficient details that others could replicate the analyses.

For instance, if you developed a questionnaire as part of this study and it is not under a copyright more restrictive than CC-BY, please include a copy, in both the original language and English, as Supporting Information.

'AP was one of the pharmacist MHFA facilitators.'

a. Please confirm that this does not alter your adherence to all PLOS ONE policies on sharing data and materials, by including the following statement: "This does not alter our adherence to  PLOS ONE policies on sharing data and materials.” (as detailed online in our guide for authors http://journals.plos.org/plosone/s/competing-interests).  If there are restrictions on sharing of data and/or materials, please state these.

Please note that we cannot proceed with consideration of your article until this information has been declared.

Reviewers' comments:

Reviewer's Responses to Questions

**Comments to the Author**

1. Is the manuscript technically sound, and do the data support the conclusions?

Reviewer #1: Partly

Reviewer #2: Yes

Reviewer #3: Partly

Reviewer #4: Partly

Reviewer #5: Partly

2. Has the statistical analysis been performed appropriately and rigorously? 

Reviewer #1: Yes

Reviewer #2: Yes

Reviewer #3: Yes

Reviewer #4: No

Reviewer #5: Yes

3. Have the authors made all data underlying the findings in their manuscript fully available?

Reviewer #1: Yes

Reviewer #2: Yes

Reviewer #3: Yes

Reviewer #4: No

Reviewer #5: Yes

4. Is the manuscript presented in an intelligible fashion and written in standard English?

Reviewer #1: Yes

Reviewer #2: Yes

Reviewer #3: Yes

Reviewer #4: Yes

Reviewer #5: Yes

5. Review Comments to the Author

Reviewer #1: Introduction, Lines 59-62: Not sure the relevance of discussing MHFA on attitudes since this was not an objective of your study. Consider deletion

Line 71: What data-informed changes do you anticipate or might result? What has the literature shown? Has previous studies shown a low uptake of MHFA or low use of skills of those who are trained? Would be nice to have a little more justification as to why this study was undertaken

Lines 95-100: Appreciate that this survey was developed from a literature search and the MHFA training assessment questions, but were any efforts undertaken to validate or pilot your survey? Are the MHFA questions validated? This is a somewhat major limitation of the study if not addressed in my opinion.

Lines 101-104: It would be nice to understand at a deeper level what your content analysis looked like and how it was coded. Where any validated frameworks used to develop these open-ended questions? If not, how were they created/validated to ensure they measured what they were designed to measure?

Table 1 -- might consider use of subheadings for each demographic o more clearly differentiate these for your readers

Lines 113-135: Might consider differentiating the reluctance items in your methods so that your reader is anticipating this. Delineating this differences and making it clear to your reader what these mean/what they were assessing

Tables 2 and 3 -- might consider presenting data as % vs numbers -- would be easier to interpret. Table 4 presents both n and % -- suggest being consistent between all tables

Lines 228--236: I think this is a little out of place and doesn't focus on the most important findings of your study and putting them into context -- an important point to highlight might be other trainings for pharmacist intervention in mental health, such as WA state where training for suicide prevention is required. Pharmacists expressed interest in continuing education/revisiting training, which is required to maintain the certification -- the MHFA is only good for 3 years. I would like to see a better analysis of what pharmacy education literature has shown as some of their findings were different than yours which is important with a significant n being student pharmacists. Additionally, Australia and Canada are much further in engaging pharmacists in this healthcare role. What can we learn from them?

Limitation-- also that the survey was not validated. Pharmacists who didn't respond might not have responded because they aren't using these skills and concerns for judgment/failing

Reviewer #2: Research into the impact of MHFA training on behavioral changes in trained pharmacists is needed. This paper provides a beginning to that evidence base. Since it was a post-training survey, the information may have been more impactful if a pre-training survey had been taken by the trainees, a consideration for future research.

Line 84: "assessing patients" This sentence is part of a paragraph discussing general application of MHFA, not specifically health care professionals. Consider whether this should be "assessing individuals" as the term "patient" implies a health care professional, not a member of the general public.

Lines 88-89: "The survey was anonymous." This sentence seems repetitive since the previous sentence states that if was an anonymous electronic survey.

Lines 91-92: Email lists. Were these email lists that were kept by the pharmacists trainers with a view to doing a study? Were these the email lists that were obtained at the start of the training that are required by MHFA for submission and certification? Was MHFA contacted about the use of these email lists for research? Were the participants who provided their email addresses at their MHFA training made aware that their email may be used in a future research study and were they given the opportunity to opt out of having their email on a research list?

Lines 161: "understating the system", should this be "understanding the system"?

Lines 200-204: This paragraph discusses how the respondents felt that they didn't know their patients well enough to be providing mental health information. Please consider discussing here how MHFA training is intended for the lay public to be able to help anyone, including a stranger or someone that they don't know well. This is a basic tenet of the training, that it is first aid that can be provided to anyone. This should be included to remind the reader that MHFA is not meant to be used only with people that the trained person knows well.

Reviewer #3: Thank you for the invitation to review the paper by Witry et al. I think this paper is adding to current literature in attempting to understand how MHFA training is used in pharmacy practice. Previous research on interventions to improve confidence and willingness to engage with people who experience mental illness tend to report pre/immediate post, self-report evaluations of the intervention and lack longitudinal follow up. There is limited research on the topic of suicide prevention in pharmacy practice or on the contribution of pharmacists beyond the dispensing of medication. While the study has many limitations as acknowledged by the authors, it is important to publish the results of this study to build evidence on the effectiveness of MHFA as a potential educational intervention for pharmacists. Below are some comments and suggestions regarding the paper that I hope the authors will find useful.

Introduction

• Title: This paper focusses on the use of MHFA as gatekeeper training for suicide or crisis management. I feel this should be reflected in the title.

• Line 19: “describe the reluctance and confidence” to do what?

• Line 45: Is the purpose of MHFA in the US to bridge gaps in access to mental health services? Perhaps the purpose of MHFA as described in the MHFA manual could be described here so that we know we are not training pharmacists as amateur counsellors.

• Line 65: The new Australian and Canadian study may be helpful to contextualise here https://ps.psychiatryonline.org/doi/10.1176/appi.ps.201900244

Methods

• How were participants recruited for the training?

• Line 83. MHFA studies usually include a slightly more detailed description of the intervention such as an overview of the content and the action plan. This would be useful to the reader as the US version is clearly different to the original Australian and European versions and most of the pharmacy research to date uses the Australian version. Did you use pharmacy specific scenarios?

• Line: Already stated the survey was anonymous.

• Line 96: The questions here seem to address attitudes to suicide prevention as much as knowledge.

• Were the questions specifically related to the use of MHFA in the pharmacy setting? Not use of the skills in a personal context or with colleague/ student? Did the question say ‘since completing MHFA have you….?

• Line 103: Was the content analysis carried out by one author alone? Quotes selected to demonstrate themes.

Results

• In general it would be useful if there were more descriptive titles in the tables so they could stand alone.

• Line 113. I find this sentence a little confusing. Perhaps clarify?

• Line 140: Any example of a success story worth sharing?

Discussion

• Line 180: I would find it more readable to simplify/clarify the language of the paper summary here.

• Line 221: The most useful addition to current literature in my opinion is the aspect on how often the pharmacists used their skills. It is a shame there was no baseline comparison.

• Line 253: Can continuing education address environmental concerns? Need to assess the work environment and identify ways of engaging with all patients privately.

Reviewer #4: Abstract:

• If room could a background be given before objectives?

• ‘4-contact design’ is unclear?

• Seems unnecessary to state ‘four emails were undeliverable in the abstract’?

• Line 33 – ‘the skill’. MHFA is about more than one skill?

Background:

• Suggest mental health ‘service’ providers

• Paragraph 3 refs 18 and 19 – sentence refers to MHFA but only one ref is about MHFA? Also there are systematic reviews and meta-analyses of effects of MHFA

• Paragraph 4 – unclear what ‘programs’ this sentence is referring to? A comprehensive discussion of current evidence base would be better.

• A concern with this paper is the discussion of ‘behaviors’ and last paragraph of introduction talks about increases in behaviors post MHFA training. It is important to note these are self-reported behaviors and rely on participants to report whether they used skills in real life or present fictitious scenarios and ask what a participant would do in these situations. It is important to note this as a limitation.

• ‘Such work also may prompt data informed changes’ – unclear what is meant by this?

Methods

• Were MHFA instructors accredited with MHFA USA?

• Reference to ref 27 on line 87 is unclear?

• Was there ethical approval to conduct this study and to email participants of the MHFA training about the study?

• I have concerns about the validity of the survey instrument. ‘Reluctance to engage’ is not a knowledge construct and appears to be more about attitudes towards providing MHFA? Can more information be provided about the psychometric properties of this scale and how it was constructed/developed?

• It is unclear what the open-ended questions were?

Results:

• Table 1 – what is P1, P2 etc?

• Not appropriate to say ‘skewed to disagreement’ or ‘skewed positive’, something needs to be statistically significant or not

• If data is skewed in one direction it is not appropriate to calculate means, it should be median, or just report percent of participants with each response.

• Suggest in tables 2 and 3 to put the labels of 1-5 in the columns to make it easier to follow

• Table 4 – says if people answer yea then they answer either ‘1, 2, 3, 4+ times’. How is a mean calculated if just 4 or more? Or was this a free response to nominate any number?

• Again – unclear on what open-ended questions were about?

Discussion:

• I feel the discussion needs some reshaping as I don’t think the claims can be appropriately backed up by the data presented in this study. There is no pre-MHFA data so it is hard to say there are high levels of confidence in MHFA skills post training when there is no baseline data. Also concerns again about he validity of the knowledge/reluctance items.

• It reports to be the first longitudinal evaluation of MHFA in pharmacy, but there is no baseline and it is unclear as to what the timeframe is that the surveys have been completed after training completion.

• Behaviors – again this should be used with care and referred to as ‘self-reported behaviors’

• Lines 217-220 – Lack of mental health services and confidence of respondents seem to be very different points? This paragraph needs some reworking

• The discussion lacks reference to a lot of work done in the space (e.g. Ashoorian et al 2018 Early Int Psych, Rodgers et al. 2019 Adv Ment Health, Boukouvalas et al. 2018 Soc Psych Psychiatr Epidemiol)

Minor:

A number of language suggestions:

• Report numbers in words when less than 10

• ‘Trainings’ – suggest training programs

• ‘Persons’ – people

• ‘mental health condition’ – suggest mental illness not condition as condition is not a diagnosed illness

• Use people living with a mental illness not patients. Mental illness can be very stigmatizing and people find it labelling to be called a ‘patient’.

Reviewer #5: Evaluation of Mental Health First Aid training initiative for pharmacy

PLOS ONE

The present manuscript is a report of a survey evaluating the outcomes of pharmacy professionals trained in MHFA. Outcomes explored include reluctance and confidence as a result of MHFA, MHFA behaviors since training, and feedback on their MHFA training.

1. Abstract: Please clarify what pharmacy professionals mean.

2. Abstract: Please indicate timing of the survey as it relates to the training.

3. Abstract: Awkward wording “conducive” responses. Please consider rewording in this section and elsewhere.

4. Introduction: Is “gatekeeper training” language typical since this language seems somewhat new and may be confusing to readers? Please clarify this language early in the paper so readers are not confused with other references to gatekeepers in healthcare.

5. Introduction: It may be helpful if authors clarify that the first aid level of care in lines 46-47 are not intended to meet the gap of the disparity in access since first aiders are not trained to provide professional-level services but rather to catch more folks missed by the system due to the disparity and that need to get into the system then managed by trained professionals. It is important to clarify this point early since most readers would be confused as to how MHFA really is addressing the gap in access to services. In some ways, it could widen the gap since there could be more folks identified by MHFA that just won’t get the needed services because there is a lack of providers, etc. Please clarify and address.

6. Introduction: Authors need to give a brief synopsis about what MHFA is, what is covered by MHFA, and how it is similar or different from other trainings. Why evaluating MHFA vs. other trainings? It should be noted that NCBH is the organization supporting MHFA training/certification. Please add.

7. Introduction: There is a small literature involving pharmacists/pharmacy students and MHFA and was surprised it was not briefly presented- at least highlighting what we do know about MHFA from these studies. Please consider adding this section.

8. Materials and Methods: It might be useful to clarify that the 5 pharmacists trained to conduct MHFA trainings were presumably trained through a week-long instructor training on how to lead MHFA trainings provided by the National Council for Behavioral Health (NCBH). Please add clarification.

9. Materials and Methods: The authors are not completely accurate that the US version is an 8-hour training. It actually can be delivered in multiple versions of two 4-hour sessions, or four 2-hr sessions. It is not clear there is a “US version”. The authors probably meant that the version their 5 pharmacists completed was 8-hour but doesn’t mean the only version available. Please clarify.

10. Materials and Methods: Please clarify when survey was administered with regards to when MHFA attendees attended their respective trainings. It would be good to note since it could be that survey was administered much further out in time from some trainings and not others. It would be useful to note if the later trainees had different perspectives from the earlier trainees. Please also indicate how many trainings were provided by each instructor? Is there any way to track responses to the trainer and see if a particular trainer had an impact vs. others? Please clarify.

11. Materials and Methods: The authors really need to provide more information about the knowledge, confidence, and use of skills items with greater details. Readers need to have a better sense of what these items measure- even examples of 2-3 items would be helpful. I would include some details in methods and refer to table for actual items. Reliability coefficients for scales should be in methods section. Please also state what types of descriptive statistics was used for which data. Please address.

12. Materials and Methods: Is knowledge truly knowledge or attitudes? Authors used an attitudinal scale of agreement and items seem more attitudes than knowledge. Please justify items as knowledge vs. attitudes and explain how agreement scale is reflection of knowledge vs. attitudes. Knowledge is usually assessed more as nominal items of correct or not.

13. Materials and Methods: Was there only one coder? Please clarify multiple coders and how discrepancies were resolved.

14. Materials and Methods: Please indicate at end of methods IRB approval.

15. Results: Table 1. Authors might want to collapse ages into fewer categories to save journal space. Also, not sure the need for P1-P4 student designations and likely lost on an international readership. It would be interesting to have collected prior experience with mental illness, etc. This might be an important to note as either a limitation or future direction. Some literature shows prior experience with mental illness either clinically or in personal lives affects attitudes and skills.

16. Results: Tables 2 and 3. Please provide n (frequency) AND % for Strongly Disagree to Strongly Agree. Most tables like this abbreviate SD for Strongly Disagree, D= Disagree, etc. See all numbers as column headers and in rows is somewhat confusing and distracting. Authors might include medians as well. Please indicate total sample size in table headers.

17. Results: Table 4. Total sample size responding to this item should be included in table header.

18. Discussion: Please consider above comments as they relate to comments in this section. Additional limitations include: generalizability of findings to other settings, lack of validated measures, lack of a comparison group and baseline measures to discern differences across time, potential temporal issues between time of trainings and survey (memory recall biases), and inability to discern trainer effects on outcomes.

19. Discussion: Authors please consider existing literature on MFHA and other trainings and relate current finding/results to such prior works. Connecting the work and results with any conceptual frameworks related to the topic would be a plus and add value to this work.

20. General comments: Authors might consider getting permission NCBH about publishing based on their program. It can’t hurt if not already done.

21. Discussion/Conclusions: Overall, these sections are generally well written with many excellent points brought up. It would been useful if the authors has a future directions section where they can propose are next steps/directions based on work and findings. They talked about future research possibilities at different points of the discussions but not as it own section with defined priorities and plans. Such a focused section based on findings and limitations will help readers see this as new program of research helping to establish the best scientific evidence for the effective use of MHFA in pharmacy practice.

This manuscript does offer some new information compared to prior publications. However, it is significant methodological limitations and could be strengthened in several areas that lack clarity and specificity. This reviewer wishes the authors the best as they pursue this work and seek publication.

6. PLOS authors have the option to publish the peer review history of their article (what does this mean?). If published, this will include your full peer review and any attached files.

Reviewer #1: No

Reviewer #2: No

Reviewer #3: No

Reviewer #4: No

Reviewer #5: No

---

## [Author Response · Author response to Decision Letter 0]

28 Feb 2020

PONE-D-19-33742

Evaluation of Mental Health First Aid training initiative for pharmacy

PLOS ONE

--I believe we have adopted the style. I did not see a web resource for file naming, but I copied what was in the video.

Please include additional information regarding the survey or questionnaire used in the study and ensure that you have provided sufficient details that others could replicate the analyses.

--We have added additional detail on the questionnaire, including a copy.

For instance, if you developed a questionnaire as part of this study and it is not under a copyright more restrictive than CC-BY, please include a copy, in both the original language and English, as Supporting Information.

--Attaching

--'AP was one of the pharmacist MHFA facilitators.'

"This does not alter our adherence to PLOS ONE policies on sharing data and materials.”

--It is probably more appropriate for this data to be available upon request. We have edited the cover letter to request this change.

 

Reviewer #1: 

Introduction, Lines 59-62: Not sure the relevance of discussing MHFA on attitudes since this was not an objective of your study. Consider deletion

--By attitudes we meant confidence and reluctance, we have revised this to be clearer.

Line 71: What data-informed changes do you anticipate or might result? What has the literature shown? Has previous studies shown a low uptake of MHFA or low use of skills of those who are trained? Would be nice to have a little more justification as to why this study was undertaken.

--Our thinking was that if a certain item had low confidence or a certain behavior seemed rare, that it could be targeted for additional training. But we have deleted this to streamline the manuscript.

Lines 95-100: Appreciate that this survey was developed from a literature search and the MHFA training assessment questions, but were any efforts undertaken to validate or pilot your survey? Are the MHFA questions validated? This is a somewhat major limitation of the study if not addressed in my opinion.

--We did get feedback from persons outside the research team from evaluation and suicidology. It is difficult to validate these questionnaires, and other authors have pointed this out when publishing their work. Suicide is fortunately relatively rare. So we tend to use instruments that have demonstrated utility in previous studies, certainly more work is needed that tests the link between attitudes and behaviors. This has been added to the discussion.

Lines 101-104: It would be nice to understand at a deeper level what your content analysis looked like and how it was coded. Where any validated frameworks used to develop these open-ended questions? If not, how were they created/validated to ensure they measured what they were designed to measure?

--This was a very basic, non-interpretive content analysis where we grouped like responses together. We have added a bit more detail including that 1 coder provided an initial sorting and another reviewed the coded data and selected representative quotes,

Table 1 -- might consider use of subheadings for each demographic o more clearly differentiate these for your readers

--Updated. Thank you.

Lines 113-135: Might consider differentiating the reluctance items in your methods so that your reader is anticipating this. Delineating this differences and making it clear to your reader what these mean/what they were assessing

--We have changed this to be more clear throughout, using reluctance instead of knowledge in most occurrences.

Tables 2 and 3 -- might consider presenting data as % vs numbers -- would be easier to interpret. Table 4 presents both n and % -- suggest being consistent between all tables

--Changed to percents only. Also added median as recommended by another reviewer

Lines 228--236: I think this is a little out of place and doesn't focus on the most important findings of your study and putting them into context -- an important point to highlight might be other trainings for pharmacist intervention in mental health, such as WA state where training for suicide prevention is required. Pharmacists expressed interest in continuing education/revisiting training, which is required to maintain the certification -- the MHFA is only good for 3 years. I would like to see a better analysis of what pharmacy education literature has shown as some of their findings were different than yours which is important with a significant n being student pharmacists. Additionally, Australia and Canada are much further in engaging pharmacists in this healthcare role. What can we learn from them?

--We have added the need for recertification at 3 years. And added a reference to Washington State and QPR. We have added more comparison to other studies elsewhere in the introduction and discussion.

Limitation-- also that the survey was not validated. Pharmacists who didn't respond might not have responded because they aren't using these skills and concerns for judgment/failing

--Thank you, we added the detail you suggested about non-response bias to the limitations

Reviewer #2: Research into the impact of MHFA training on behavioral changes in trained pharmacists is needed. This paper provides a beginning to that evidence base. Since it was a post-training survey, the information may have been more impactful if a pre-training survey had been taken by the trainees, a consideration for future research.

--Agreed, we have this in future research and limitations.

Line 84: "assessing patients" This sentence is part of a paragraph discussing general application of MHFA, not specifically health care professionals. Consider whether this should be "assessing individuals" as the term "patient" implies a health care professional, not a member of the general public.

--Good catch, thank you, we have made this change.

Lines 88-89: "The survey was anonymous." This sentence seems repetitive since the previous sentence states that if was an anonymous electronic survey.

--Deleted

Lines 91-92: Email lists. Were these email lists that were kept by the pharmacists trainers with a view to doing a study? Were these the email lists that were obtained at the start of the training that are required by MHFA for submission and certification? Was MHFA contacted about the use of these email lists for research? Were the participants who provided their email addresses at their MHFA training made aware that their email may be used in a future research study and were they given the opportunity to opt out of having their email on a research list?

--We have added details about opting out and where the email addresses came from. We did discuss or research project with MHFA USA and were encouraged to proceed. IRB has been added and details about protecting human subjects has been added.

Lines 161: "understating the system", should this be "understanding the system"?

--Thank you for catching this. Changed

Lines 200-204: This paragraph discusses how the respondents felt that they didn't know their patients well enough to be providing mental health information. Please consider discussing here how MHFA training is intended for the lay public to be able to help anyone, including a stranger or someone that they don't know well. This is a basic tenet of the training, that it is first aid that can be provided to anyone. This should be included to remind the reader that MHFA is not meant to be used only with people that the trained person knows well.

--This is a good point and we have added “However, MHFA is intended for any person, regardless of background, to help someone without needing a prior relationship.( 6)”

Reviewer #3: Thank you for the invitation to review the paper by Witry et al. I think this paper is adding to current literature in attempting to understand how MHFA training is used in pharmacy practice. Previous research on interventions to improve confidence and willingness to engage with people who experience mental illness tend to report pre/immediate post, self-report evaluations of the intervention and lack longitudinal follow up. There is limited research on the topic of suicide prevention in pharmacy practice or on the contribution of pharmacists beyond the dispensing of medication. While the study has many limitations as acknowledged by the authors, it is important to publish the results of this study to build evidence on the effectiveness of MHFA as a potential educational intervention for pharmacists. Below are some comments and suggestions regarding the paper that I hope the authors will find useful.

--Thank you for your review

Introduction

Title: This paper focusses on the use of MHFA as gatekeeper training for suicide or crisis management. I feel this should be reflected in the title.

--Title has been updated to better reflect content.

Line 19: “describe the reluctance and confidence” to do what?

--We have added “to intervene in mental health crises” to clarify. 

Line 45: Is the purpose of MHFA in the US to bridge gaps in access to mental health services? Perhaps the purpose of MHFA as described in the MHFA manual could be described here so that we know we are not training pharmacists as amateur counsellors.

--Great recommendation, we have added this line to the end of the first paragraph. Mental Health First Aid is one such training with the goal of training community members to serve as a “first line of support” for distressed persons and helping the person in crisis seek further assistance.

Line 65: The new Australian and Canadian study may be helpful to contextualise here https://ps.psychiatryonline.org/doi/10.1176/appi.ps.201900244

--Thank you, we have added to the introduction and discussion.

Methods

How were participants recruited for the training?

---Recruitment varied, we added some detail to the methods. The trainings were offered through a state pharmacy association to local individuals via an online signup, to student pharmacist groups, and to workshop attendees,

Line 83. MHFA studies usually include a slightly more detailed description of the intervention such as an overview of the content and the action plan. This would be useful to the reader as the US version is clearly different to the original Australian and European versions and most of the pharmacy research to date uses the Australian version. Did you use pharmacy specific scenarios?

---We have added additional detail, thank you for the recommendation. Mental Health First Aid is one such training with the goal of training community members to serve as a “first line of support” for distressed persons and helping the person in crisis seek further assistance.( 6) Mental Health First Aid U.S.A. is based on a 5-step action plan that uses the acronym ALGEE®: assess for risk of suicide or self-harm, listen non-judgmentally, give reassurance and information, encourage appropriate professional help, and encourage self-help and other support strategies.(6)

Line: Already stated the survey was anonymous.

--Deleted.

Line 96: The questions here seem to address attitudes to suicide prevention as much as knowledge.

--We have edited this and are consistently referring to “reluctance” instead of knowledge. This is recommended by the RAND article.

Were the questions specifically related to the use of MHFA in the pharmacy setting? Not use of the skills in a personal context or with colleague/ student? 

--We chose to be general and have added this statement to the methods. Survey questions were phrased generally as opposed to referring specifically to the respondent’s professional role because as a gateleeper training, the goal is to give persons skills to use regardless of the setting. We have added this clarification to the discussion.

Did the question say ‘since completing MHFA have you….?

--We have attached the survey. The survey contained this prompt prior to asking about behaviors. Please report your best estimate of the number of times that you have had experience with the situation since completing MHFA training. 

 Line 103: Was the content analysis carried out by one author alone? Quotes selected to demonstrate themes.

--Two authors worked on the content analysis. We are hesitant to call these themes given it’s a content analysis.

Results

In general it would be useful if there were more descriptive titles in the tables so they could stand alone.

--Thank you, we have made the tables more descriptive.

Line 113. I find this sentence a little confusing. Perhaps clarify?

--We have reworded all uses of conducive per other reviewer. This should help clarify.

Line 140: Any example of a success story worth sharing?

---Added

Discussion

Line 180: I would find it more readable to simplify/clarify the language of the paper summary here.

--We have clarified, removing conducive throughout.

Line 221: The most useful addition to current literature in my opinion is the aspect on how often the pharmacists used their skills. It is a shame there was no baseline comparison.

--Agreed.

Line 253: Can continuing education address environmental concerns? Need to assess the work environment and identify ways of engaging with all patients privately.

--We have added more detail to the body.

Reviewer #4: 

Abstract: If room could a background be given before objectives?

--Added

‘4-contact design’ is unclear?

--We have edited this throughout to 4 email sequence.

Seems unnecessary to state ‘four emails were undeliverable in the abstract’?

--deleted

Line 33 – ‘the skill’. MHFA is about more than one skill?

--This has been clarified.

Background:

Suggest mental health ‘service’ providers

--changed

Paragraph 3 refs 18 and 19 – sentence refers to MHFA but only one ref is about MHFA? Also there are systematic reviews and meta-analyses of effects of MHFA

--Added, thank you for the suggestion,

Paragraph 4 – unclear what ‘programs’ this sentence is referring to? A comprehensive discussion of current evidence base would be better.

--This paragraph has been rewritten to address this concern.

A concern with this paper is the discussion of ‘behaviors’ and last paragraph of introduction talks about increases in behaviors post MHFA training. It is important to note these are self-reported behaviors and rely on participants to report whether they used skills in real life or present fictitious scenarios and ask what a participant would do in these situations. It is important to note this as a limitation.

--Agreed. We have emphasized this more as a limitation. 

‘Such work also may prompt data informed changes’ – unclear what is meant by this?

---We have clarified this. Also, variation in confidence, reluctance, and self-reported behaviors could prompt supplemental education.(

Methods

Were MHFA instructors accredited with MHFA USA?

--Yes, we have added this clarification.

Reference to ref 27 on line 87 is unclear?

--The evaluation is available from the group that conducted it. 

Was there ethical approval to conduct this study and to email participants of the MHFA training about the study?

--Yes we have added the IRB information and detail about how persons could opt out. 

I have concerns about the validity of the survey instrument. ‘Reluctance to engage’ is not a knowledge construct and appears to be more about attitudes towards providing MHFA? Can more information be provided about the psychometric properties of this scale and how it was constructed/developed?

--We based this on the RAND gatekeeper review study and have clarified by removing reference to knowledge. We provide 2 references for the reluctance items. The Cronbach alphas are provided underneath the two tables. These items we adapted from other articles.

It is unclear what the open-ended questions were?

--Thanks for pointing this out. We have added them to the methods.

Results:

Table 1 – what is P1, P2 etc?

--Year in the pharmacy program, we have deleted and just say pharmacy students 

Not appropriate to say ‘skewed to disagreement’ or ‘skewed positive’, something needs to be statistically significant or not

---We have removed the term skew given its statistical meaning.

If data is skewed in one direction it is not appropriate to calculate means, it should be median, or just report percent of participants with each response.

--We have removed mean and Std dev in favor of median and IQR.

Suggest in tables 2 and 3 to put the labels of 1-5 in the columns to make it easier to follow

--Added, thank you for the recommendation.

Table 4 – says if people answer yea then they answer either ‘1, 2, 3, 4+ times’. How is a mean calculated if just 4 or more? Or was this a free response to nominate any number?

--We have clarified in the methods and are presenting a median and IQR>

Again – unclear on what open-ended questions were about?

--Added to the methods.

Discussion:

I feel the discussion needs some reshaping as I don’t think the claims can be appropriately backed up by the data presented in this study. There is no pre-MHFA data so it is hard to say there are high levels of confidence in MHFA skills post training when there is no baseline data. Also concerns again about he validity of the knowledge/reluctance items.

--We have made changes to the discussion and point out in the limitations the lack of pre data.

It reports to be the first longitudinal evaluation of MHFA in pharmacy, but there is no baseline and it is unclear as to what the timeframe is that the surveys have been completed after training completion.

---We have calculated the timeframe of between 6 and 18 months since training and added that to the results.

Behaviors – again this should be used with care and referred to as ‘self-reported behaviors’

--- We have added this throughout. 

Lines 217-220 – Lack of mental health services and confidence of respondents seem to be very different points? This paragraph needs some reworking

---We have moved this statement out and put with other open ended items in a subsequent paragraph, and added additional context.

The discussion lacks reference to a lot of work done in the space (e.g. Ashoorian et al 2018 Early Int Psych, Rodgers et al. 2019 Adv Ment Health, Boukouvalas et al. 2018 Soc Psych Psychiatr Epidemiol)

---Thank you for these references,. We have worked to incorporate,

Minor:

A number of language suggestions:

• Report numbers in words when less than 10 - changed

• ‘Trainings’ – suggest training programs - changed

• ‘Persons’ – people - changed

• ‘mental health condition’ – suggest mental illness not condition as condition is not a diagnosed illness - changed

• Use people living with a mental illness not patients. Mental illness can be very stigmatizing and people find it labelling to be called a ‘patient’. Changed most occurrences. Pharmacists may call their patrons patients, especially if they are in a service role.

---All adopted, thank you.

Reviewer #5: Evaluation of Mental Health First Aid training initiative for pharmacy

PLOS ONE

The present manuscript is a report of a survey evaluating the outcomes of pharmacy professionals trained in MHFA. Outcomes explored include reluctance and confidence as a result of MHFA, MHFA behaviors since training, and feedback on their MHFA training.

Abstract: Please clarify what pharmacy professionals mean.

--pharmacists, technicians, students, - clarified.

Abstract: Please indicate timing of the survey as it relates to the training.

---We do say that it was administered in May and June 2019 and the training sessions occurred in 2018, Changed during to throughout,

Abstract: Awkward wording “conducive” responses. Please consider rewording in this section and elsewhere.

---Conducive has been eliminated throughout.

Introduction: Is “gatekeeper training” language typical since this language seems somewhat new and may be confusing to readers? Please clarify this language early in the paper so readers are not confused with other references to gatekeepers in healthcare.

---We have introduced this term earlier in a revised introduction.

Introduction: It may be helpful if authors clarify that the first aid level of care in lines 46-47 are not intended to meet the gap of the disparity in access since first aiders are not trained to provide professional-level services but rather to catch more folks missed by the system due to the disparity and that need to get into the system then managed by trained professionals. It is important to clarify this point early since most readers would be confused as to how MHFA really is addressing the gap in access to services. In some ways, it could widen the gap since there could be more folks identified by MHFA that just won’t get the needed services because there is a lack of providers, etc. Please clarify and address.

---We have added a new paragraph about MHFA, but feel it is beyond the scope of this paper to raise the concern that making referrals will be a burden on the mental health care infrastructure.

Introduction: Authors need to give a brief synopsis about what MHFA is, what is covered by MHFA, and how it is similar or different from other trainings. Why evaluating MHFA vs. other trainings? It should be noted that NCBH is the organization supporting MHFA training/certification. Please add.

---Added

Introduction: There is a small literature involving pharmacists/pharmacy students and MHFA and was surprised it was not briefly presented- at least highlighting what we do know about MHFA from these studies. Please consider adding this section.

----We have added more pharmacy studies to the introduction.

Materials and Methods: It might be useful to clarify that the 5 pharmacists trained to conduct MHFA trainings were presumably trained through a week-long instructor training on how to lead MHFA trainings provided by the National Council for Behavioral Health (NCBH). Please add clarification.

---added

Materials and Methods: The authors are not completely accurate that the US version is an 8-hour training. It actually can be delivered in multiple versions of two 4-hour sessions, or four 2-hr sessions. It is not clear there is a “US version”. The authors probably meant that the version their 5 pharmacists completed was 8-hour but doesn’t mean the only version available. Please clarify.

---We have clarified that sessions were either 1 8 hour session or 2 4 hour sessions..

Materials and Methods: Please clarify when survey was administered with regards to when MHFA attendees attended their respective trainings. It would be good to note since it could be that survey was administered much further out in time from some trainings and not others. It would be useful to note if the later trainees had different perspectives from the earlier trainees. Please also indicate how many trainings were provided by each instructor? Is there any way to track responses to the trainer and see if a particular trainer had an impact vs. others? Please clarify.

----We have added that the survey was administered between 6 and 18 months following the trainings. We have decided not to do sub-analyses based on time since training because there would be too many confounding variables. Because the survey was anonymous, we cannot tie participant responses to an individual trainer. We have added to the limitations “Also, it was not possible to evaluate differences among trainers nor were adjustments made based on the gap between training and survey completion.”

Materials and Methods: The authors really need to provide more information about the knowledge, confidence, and use of skills items with greater details. Readers need to have a better sense of what these items measure- even examples of 2-3 items would be helpful. I would include some details in methods and refer to table for actual items. Reliability coefficients for scales should be in methods section. Please also state what types of descriptive statistics was used for which data. Please address.

---We have added example items and their corresponding concepts. We have included “Descriptive statistics were calculated for all scaled and multiple-choice responses.”

Materials and Methods: Is knowledge truly knowledge or attitudes? Authors used an attitudinal scale of agreement and items seem more attitudes than knowledge. Please justify items as knowledge vs. attitudes and explain how agreement scale is reflection of knowledge vs. attitudes. Knowledge is usually assessed more as nominal items of correct or not.

---We changed knowledge to reluctance which should address this.. 

Materials and Methods: Was there only one coder? Please clarify multiple coders and how discrepancies were resolved.

---Added description of coding and discussions to methods

Materials and Methods: Please indicate at end of methods IRB approval.

---Added at the beginning of methods.

Results: Table 1. Authors might want to collapse ages into fewer categories to save journal space. Also, not sure the need for P1-P4 student designations and likely lost on an international readership. It would be interesting to have collected prior experience with mental illness, etc. This might be an important to note as either a limitation or future direction. Some literature shows prior experience with mental illness either clinically or in personal lives affects attitudes and skills.

---year in pharmacy school deleted

Results: Tables 2 and 3. Please provide n (frequency) AND % for Strongly Disagree to Strongly Agree. Most tables like this abbreviate SD for Strongly Disagree, D= Disagree, etc. See all numbers as column headers and in rows is somewhat confusing and distracting. Authors might include medians as well. Please indicate total sample size in table headers.

We have changed to median and IQR. We have clarified the tables. Added sample size to table headers. Other reviewers have recommended just using % since the n is so close to 100.

Results: Table 4. Total sample size responding to this item should be included in table header.

---Done

Discussion: Please consider above comments as they relate to comments in this section. Additional limitations include: generalizability of findings to other settings, lack of validated measures, lack of a comparison group and baseline measures to discern differences across time, potential temporal issues between time of trainings and survey (memory recall biases), and inability to discern trainer effects on outcomes.

---We have added to the limitations

Discussion: Authors please consider existing literature on MFHA and other trainings and relate current finding/results to such prior works. Connecting the work and results with any conceptual frameworks related to the topic would be a plus and add value to this work.

---Added additional focus on Burnette RAND model

General comments: Authors might consider getting permission NCBH about publishing based on their program. It can’t hurt if not already done.

----We already have been in contact.

Discussion/Conclusions: Overall, these sections are generally well written with many excellent points brought up. It would been useful if the authors has a future directions section where they can propose are next steps/directions based on work and findings. They talked about future research possibilities at different points of the discussions but not as it own section with defined priorities and plans. Such a focused section based on findings and limitations will help readers see this as new program of research helping to establish the best scientific evidence for the effective use of MHFA in pharmacy practice.

----We have added future research section and removed from other places in the discussion. 

This manuscript does offer some new information compared to prior publications. However, it is significant methodological limitations and could be strengthened in several areas that lack clarity and specificity. This reviewer wishes the authors the best as they pursue this work and seek publication.

---Thank you, we have aggregated the future research ideas into its own section following limitations.

---

## [Decision Letter · Decision Letter 1]

14 Apr 2020

PONE-D-19-33742R1

Evaluation of participant reluctance, confidence, and self-reported behaviors since being trained in a pharmacy Mental Health First Aid initiative

PLOS ONE

Dear Dr Witry,

Thank you for submitting your manuscript to PLOS ONE. After careful consideration, we feel that it has merit but does not fully meet PLOS ONE’s publication criteria as it currently stands. Therefore, we invite you to submit a revised version of the manuscript that addresses the points raised during the review process.

Please consider the comments from reviewer 2 and acknowledge any limitations not already included and resubmit.

We would appreciate receiving your revised manuscript by 28 April 2020. To enhance the reproducibility of your results, we recommend that if applicable you deposit your laboratory protocols in protocols.io, where a protocol can be assigned its own identifier (DOI) such that it can be cited independently in the future. For instructions see: http://journals.plos.org/plosone/s/submission-guidelines#loc-laboratory-protocols

We look forward to receiving your revised manuscript.

Kind regards,

Andrew Soundy

Academic Editor

PLOS ONE

Reviewers' comments:

Reviewer's Responses to Questions

**Comments to the Author**

1. If the authors have adequately addressed your comments raised in a previous round of review and you feel that this manuscript is now acceptable for publication, you may indicate that here to bypass the “Comments to the Author” section, enter your conflict of interest statement in the “Confidential to Editor” section, and submit your "Accept" recommendation.

Reviewer #1: (No Response)

Reviewer #2: All comments have been addressed

Reviewer #5: All comments have been addressed

2. Is the manuscript technically sound, and do the data support the conclusions?

Reviewer #1: Partly

Reviewer #2: Yes

Reviewer #5: Yes

3. Has the statistical analysis been performed appropriately and rigorously? 

Reviewer #1: Yes

Reviewer #2: Yes

Reviewer #5: Yes

4. Have the authors made all data underlying the findings in their manuscript fully available?

Reviewer #1: Yes

Reviewer #2: Yes

Reviewer #5: (No Response)

5. Is the manuscript presented in an intelligible fashion and written in standard English?

Reviewer #1: Yes

Reviewer #2: Yes

Reviewer #5: Yes

6. Review Comments to the Author

Reviewer #1: Thank you for your work to address the comments made by your reviewers. Your time, consideration of these requests and efforts are greatly appreciated. However, based on my concerns and concerns that were raised by at least two other reviewers regarding the validity of your survey tool/findings, I cannot recommend publication. While MHFA and other mental health roles are growing in community pharmacy practice and it is important to understand the impact of these trainings, I feel the analysis is too focused on self-reported post training and did not include a baseline assessment that hinders its findings. My greatest concern was that a validated survey tool was not used -- there are several validated attitude scales regarding suicide (ATTS, Understanding of Suicidal Patients Scale) that could have been used. Further, a qualitative framework could have been used (such as the Theory of Planned Behavior) if you felt that the validated surveys were not a good fit.

Additional comments:

1) LIne 71 -- suggest using language suggested by APhA community-based

2) Suggest including reference to the survey in your methods if intended to be included in your published article

3) Domains unclear (Line 120). Only mention demographics and then .  suggest listing all domains and then moving into description of each

4) Time since training being 6-18 months. I feel the survey not being consistently conducted xx amount of time since being completed is a significant limitation as this could impact retention of concepts, how much they are using, etc

5) Relevance of data in lines 234-239 since the national behavorial council created the training and is concerned with fidelity and does not allow adjustments to training?

Reviewer #2: All comments have been appropriately addressed, appreciate author responses to reviewer recommendations.

Reviewer #5: I think the authors addressed my primary concerns. I will say there may be still some typos like line 130 "participants". I am still not entirely convinced how knowledge is reluctance but will accept their attempt here. I might have suggested that they just refer to reluctance and not view it as knowledge at all. Further, I didn't see anywhere why the focus on MHFA over other trainings like MHFA.

7. PLOS authors have the option to publish the peer review history of their article (what does this mean?). If published, this will include your full peer review and any attached files.

Reviewer #1: No

Reviewer #2: Yes: Carol A. Ott, PharmD, BCPP

Reviewer #5: No

---

## [Author Response · Author response to Decision Letter 1]

14 Apr 2020

Please see attached response to reviewers document

---

## [Editor Report · Decision Letter 2]

20 Apr 2020

Evaluation of participant reluctance, confidence, and self-reported behaviors since being trained in a pharmacy Mental Health First Aid initiative

PONE-D-19-33742R2

Dear Dr. Witry,

We are pleased to inform you that your manuscript has been judged scientifically suitable for publication and will be formally accepted for publication once it complies with all outstanding technical requirements.

With kind regards,

Andrew Soundy

Academic Editor

PLOS ONE
---

## [Editor Report · Acceptance letter]

23 Apr 2020

PONE-D-19-33742R2 

Evaluation of participant reluctance, confidence, and self-reported behaviors since being trained in a pharmacy Mental Health First Aid initiative 

Dear Dr. Witry:

I am pleased to inform you that your manuscript has been deemed suitable for publication in PLOS ONE. Congratulations! Your manuscript is now with our production department. 

With kind regards,

on behalf of

Dr. Andrew Soundy 

Academic Editor

PLOS ONE